# Bacterially derived synthetic mimetics of mammalian oligomannose prime antibody responses that neutralize HIV infectivity

Ralph Pantophlet [1,2,3], Nino Trattnig[4], Sasha Murrell[5,6], Naiomi Lu[1], Dennis Chau[1], Caitlin Rempel[1], Ian A. Wilson [5,6,7,8] & Paul Kosma [4]

Oligomannose-type glycans are among the major targets on the gp120 component of the HIV envelope protein (Env) for broadly neutralizing antibodies (bnAbs). However, attempts to elicit oligomannose-specific nAbs by immunizing with natural or synthetic oligomannose have so far not been successful, possibly due to B cell tolerance checkpoints. Here we design and synthesize oligomannose mimetics, based on the unique chemical structure of a recently identified bacterial lipooligosaccharide, to appear foreign to the immune system. One of these mimetics is bound avidly by members of a family of oligomannose-specific bnAbs and their putative common germline precursor when presented as a glycoconjugate. The crystal structure of one of the mimetics bound to a member of this bnAb family confirms the antigenic resemblance. Lastly, immunization of human-antibody transgenic animals with a lead mimetic evokes nAbs with specificities approaching those of existing bnAbs. These results provide evidence for utilizing antigenic mimicry to elicit oligomannose-specific bnAbs to HIV-1.

[1] Faculty of Health Sciences, Simon Fraser University, Burnaby, BC, Canada V5A1S6. [2] Department of Molecular Biology & Biochemistry, Simon Fraser University, Burnaby, BC, Canada V5A1S6. [3] SFU Interdisciplinary Research Centre for HIV, Simon Fraser University, Burnaby, BC, Canada V5A1S6. [4] Department of Chemistry, University of Natural Resources and Life Sciences, A-1190 Vienna, Austria. [5] Department of Integrative Structural and Computational Biology, The Scripps Research Institute, La Jolla, CA 92037, USA. [6] Skaggs Institute for Chemical Biology, The Scripps Research Institute, La Jolla, CA 92037, USA. [7] IAVI Neutralizing Antibody Center, The Scripps Research Institute, La Jolla, CA 92037, USA. [8] Center for HIV/AIDS Vaccine Immunology and Immunogen Discovery, The Scripps Research Institute, La Jolla, CA 92037, USA. Correspondence and requests for materials should be addressed to R.P. (email: rpantophlet@sfu.ca) or to I.A.W. (email: wilson@scripps.edu) or to P.K. (email: paul.kosma@boku.ac.at)

The isolation of broadly neutralizing antibodies (bnAbs) from many different HIV-infected individuals over the last several years has substantially bolstered efforts to develop immunogens that might elicit similar antibodies[1]. Among the specificities that perhaps have garnered the greatest interest are bnAbs targeting a conserved patch of oligomannose-type glycans on the gp120 outer domain of the HIV-1 envelope spike (Env)[2]. Representatives of these types of antibodies typically exhibit potent neutralizing activity, which has translated to protection against robust viral challenge in relevant animal models at even modest serum titers[3] and provided a strong impetus for exploring strategies to elicit equivalent nAbs.

However, despite substantial effort, attempts to elicit oligomannose-specific antibodies with capacity to effectively neutralize HIV have not been successful[4–6]. One possible explanation for the limited success so far is restrictions imposed by B cell tolerance mechanisms. Indeed, the decoration of viral pathogens, such as HIV, with self-glycans is believed to enable immune evasion[5]. Although extensive clustering of oligomannose-type glycans on HIV-1 is rare on human epithelia[7], some human plasma glycoproteins do sparsely express oligomannose-type glycans under normal physiological conditions[7]. The occurrence of these glycans, even though not abundant, may be sufficient to limit the frequency of naïve B cells with receptors able to bind oligomannose or render such 'self-reactive' B cells anergic[8]. In that context, it is noteworthy that a high incidence of seemingly autoreactive antibodies has been observed in many HIV-infected subjects who develop bnAbs[9, 10]. Although it is not yet clear whether the development of such autoreactive antibodies correlates with certain bnAb specificities, tolerance mechanisms have been shown to limit the development of B cells expressing the mature or germline (gl) sequences of certain bnAbs in knockin mice[11–13].

Immunological tolerance, if indeed limiting the frequency or development of B cells with the requisite oligomannose specificity, might be overcome with immunogens designed in a manner that allows anergic or naive B cells with the desired specificity to be stimulated. One potential way to achieve this for carbohydrate antigens is through antigenic mimicry. It is well-established that antigenic mimicry of mammalian host structures, such as by bacterial lipooligosaccharides (LOS), can result in antibodies that are cross-reactive with host glycans. Mimicry of gangliosides by LOS of *Campylobacter jejuni* strains[14], neural cell adhesion molecules by LOS of *Neisseria meningitidis* group B[15], and Lewis blood group antigens by LOS of *Helicobacter pylori* strains[16] are perhaps the best known natural examples of this phenomenon. The elicitation of antibodies that cross-react with host glycans can also be achieved artificially; for example, heterologous presentation of human self-polysaccharides in an unnatural context evokes fairly robust antibody titers to these otherwise poorly immunogenic glycans[17]. Thus, antigenic mimicry, in the 'proper' foreign context, can overcome immune tolerance.

We reported not long ago on the unique chemical structure of the LOS from the Gram-negative plant bacterium *Rhizobium radiobacter* Rv3, comprised of a tetra-mannose backbone segment that is analogous to the D1 arm of mammalian oligomannose[18] (Fig. 1a). Using nAb 2G12, the Rv3 oligosaccharide backbone was shown to be antigenically equivalent to the D1 arm of oligomannose[18], for which the antibody is specific. The antigenic equivalency of the Rv3 backbone and the D1 arm was conclusively demonstrated from the crystal structure of 2G12 in complex with a fragment of the Rv3 oligosaccharide encompassing the D1-like segment[19]. Based on the structure complex, we posited that synthetic derivatives of the Rv3 oligosaccharide might be designed to more fully mimic mammalian oligomannose and, consequently, more readily elicit cross-reactive antibodies.

Here we report on the successful synthesis of a first panel of Rv3 oligosaccharide derivatives. From this panel, we identified a lead mimetic that is bound by members of the PGT128 or PGT130 antibody families[20] as well as their putative common gl precursor. PGT128 and related antibodies bind conserved glycans within the oligomannose patch on HIV-1 and exhibit broad neutralizing activity, and thus are of significance to vaccine design efforts. We further present the crystal structure at 2.3 Å resolution of PGT128 in complex with the lead mimetic and illustrate that, as a neoglycoconjugate, the mimetic can prime for antibodies

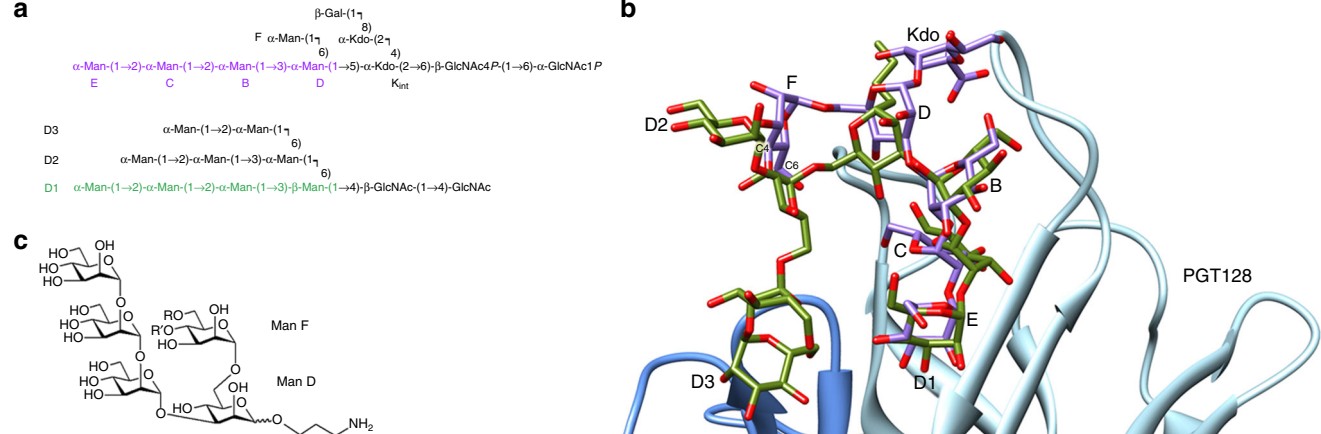

**Fig. 1** Analogy between the carbohydrate backbone of *R. radiobacter* Rv3 LOS and the D1 arm of oligomannose and conceptual design of Rv3 oligosaccharide derivatives. **a** Chemical structure of the carbohydrate backbone of Rv3 LOS (top), highlighting the tetra-mannosyl sequence of units E-C-B-D (purple) that is analogous to the chemical structure of the D1 arm (green) of Man₉ (bottom). The two sequences differ in the anomeric configuration of the first branched mannosyl unit, which is α in the Rv3 oligosaccharide (unit D) and β in oligomannose. **b** Complex of bnAb PGT128 (heavy chain: light blue, light chain: dark blue) and Man₉ (green) (PDB 3TV3) with the crystallized fragment of the Rv3 oligosaccharide (purple) (PDB 4RBP) modeled into the PGT128-binding site using the D1 arm of Man₉ as a guide. The modeling suggested that a mannosyl extension at position C4 or C6 of unit F might form a structural surrogate of the D3 arm. **c** Core pentasaccharide scaffold Man-(1 → 2)-Man-(1 → 2)-Man-(1 → 3)-[R-Man-(1 → 6)]-Man-(1 → Spacer), derived from the chemical structure of the crystallized Rv3 oligosaccharide fragment, designed to incorporate, based on modeling of the Rv3 oligosaccharide into the PGT128-binding site as in **b**, extensions at the side-chain mannose unit F at residues R (position C6) and R' (position C4). Spacer = O(CH₂)₃NH₂

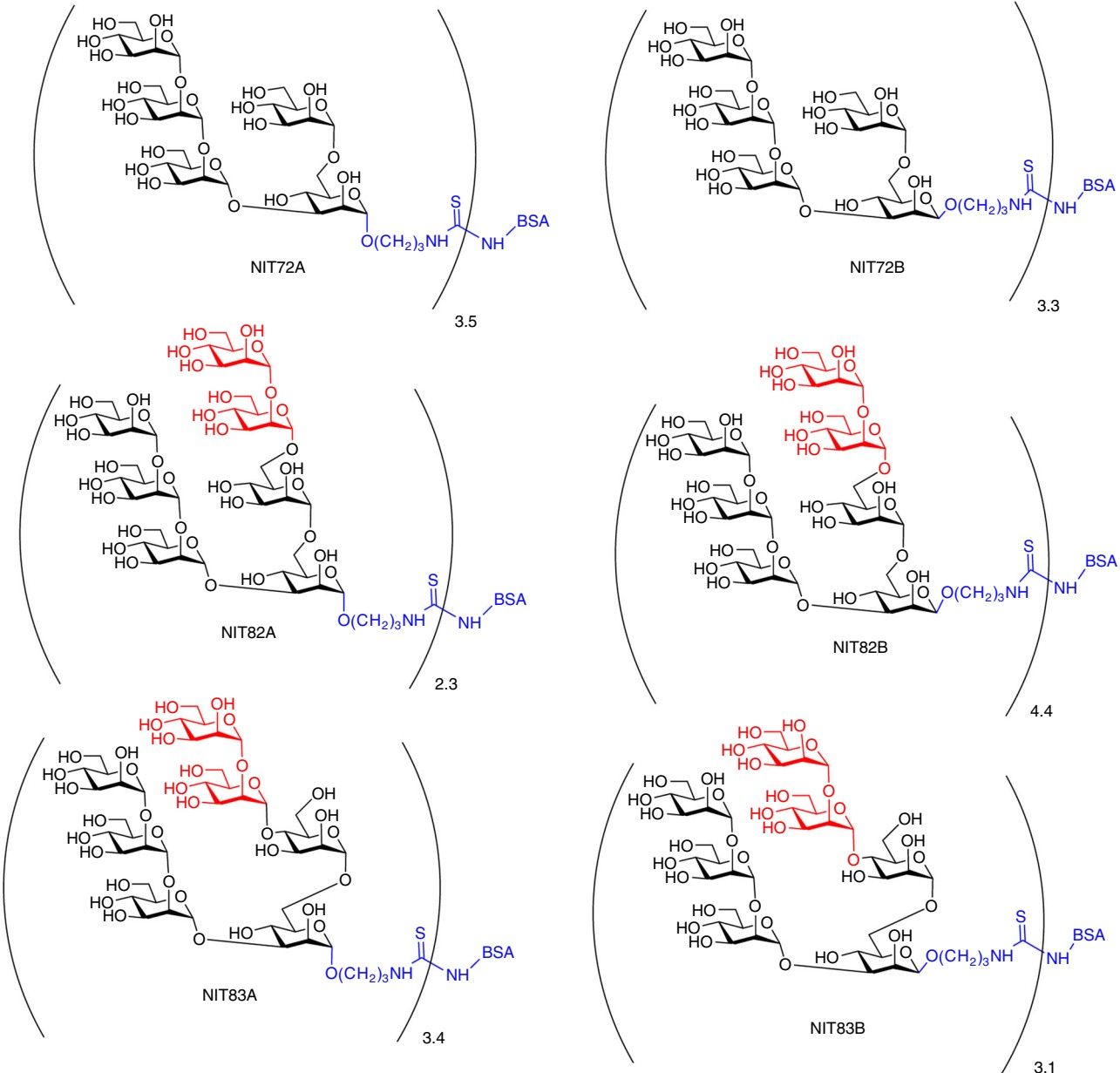

**Fig. 2** Rv3-derived neoglycoconjugates. Synthetic versions of Rv3 and Rv3 derivatives incorporating a D3-like arm were coupled to BSA via lysine residues to generate neoglycoconjugates. The average number of ligand molecules per BSA for each conjugate is shown beside each bracket and was determined by MALDI-TOF (Supplementary Figs 27-32). NIT72A and NIT72B are BSA conjugates with α and β anomers, respectively, of synthetic Rv3 oligosaccharide. NIT82A and NIT82B are BSA conjugates of synthetic derivatives (α and β anomers, respectively) of the Rv3 oligosaccharide with an α1-6-linked D3-like arm extension, based on the model of the Rv3 oligosaccharide in complex with PGT128 (Fig. 1b). NIT83A and NIT83B are BSA conjugates carrying synthetic derivatives of the Rv3 oligosaccharide (α and β anomers, respectively) with an α1-4-linked D3-like arm extension, as suggested from modeling of the Rv3 oligosaccharide in the PGT128-binding site (Fig. 1b)

with HIV-neutralizing activity in transgenic rats harboring an unarranged human immunoglobulin (Ig) repertoire.

## Results

**Structure-aided design of Rv3 oligosaccharide derivatives**. Our conceptual design of Rv3 oligosaccharide derivatives began by first modeling the structure of the crystallized Rv3 oligosaccharide fragment[19] onto that of Man$_9$ oligomannose in complex with bnAb PGT128 (PDB 3TV3; Fig. 1b). We focused on incorporating a D3-arm-like extension because the D3 arm, along with the D1 arm, are known to be important for the interaction of PGT128

and related bnAbs with oligomannose[21]. The modeling suggested that an extension at either the C-4 or C-6 position of the side-chain mannose unit of the Rv3 oligosaccharide (Man F unit; Fig. 1a, b) would likely mimic the D3 arm of oligomannose.

On the basis of the modeling and the Rv3 oligosaccharide structure, we designed a general target structure from which to synthesize Rv3 derivatives (Fig. 1c). We then synthesized two mannopentaosides (NIT59A and NIT59B) equivalent to the Rv3 oligosaccharide backbone and a set of four mannoheptaosides (NIT68A, NIT68B, NIT70A, and NIT70B) that incorporate D3-arm-like extensions. Synthesis was achieved over 12–13 steps in good to excellent yields from multi-gram scale produced mannose

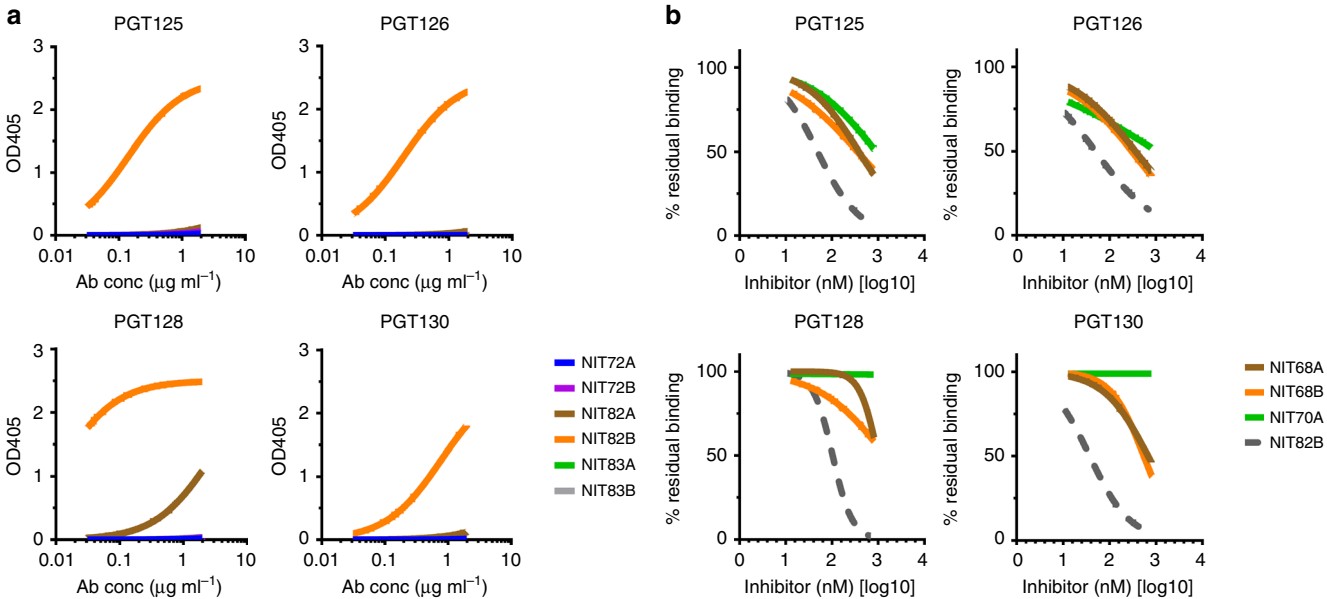

**Fig. 3** NAbs PGT125, 126, 128, and 130 bind avidly to NIT82B, a neoglycoconjugate presenting an Rv3-derived mimic of oligomannose. **a** The listed conjugates are those from Fig. 2. The conjugates (69–72 kDa) were coated as solid-phase antigen onto ELISA plate wells at 5 µg ml⁻¹ and assayed for recognition by PGT125, 126, 128, and 130 (starting at 2 µg ml⁻¹). All antibodies were tested as IgGs. The data represent the means from two independent experiments, with error bars omitted for clarity. GraphPad Prism (version 7.02) was used to determined EC$_{50}$ values (0.1–5 nM). **b** To assess the importance of valency for antibody binding, conjugate NIT82B was coated onto ELISA plates (5 µg ml⁻¹) and antibody binding assessed in the presence of the unconjugated 3-aminopropyl glycosides NIT68A (presented on NIT82A), NIT68B (presented on NIT82B) or NIT70A (presented on NIT83A), or the NIT82B glycoconjugate as a comparator (dashed line). The results, obtained from one experiment, show that the antibodies are inhibited best by the NIT82B conjugate

donors and acceptors (Supplementary Methods and Supplementary Figs 1–4). We purposely made our designs without a D2-like arm, which we reasoned would increase the likelihood that these mimetics would be more readily sensed as 'foreign' by the immune system. All structures were verified by NMR (Supplementary Figs 5–26).

**Identification of a lead oligomannose mimetic.** We generated six neoglycoconjugates of each of the oligomannosides described above by conjugation onto BSA (Fig. 2). The resulting glycoconjugates were given new name designations so as to differentiate them from their soluble counterparts: NIT72A and NIT72B, carrying glycosides equivalent to the Rv3 oligosaccharide backbone; NIT82A and NIT82B, carrying Rv3 derivatives with incorporation of a 1-6-linked D3-like extension; and NIT83A and NIT83B, carrying an Rv3 derivative with a 1-4-linked D3-like extension. The 'A' and 'B' in the name designations of the neoglycoconjugates correspond to the anomeric configuration of the mannosyl residue (α or β, respectively) at the reducing end of the glycoside (Man D unit; Fig. 1). All oligomannosides were conjugated to primary amines on BSA by activating the terminal amino group of the spacer (O(CH₂)₃NH₂) located at the reducing end of the glycoside as the isothiocyanate. A low conjugation density was chosen to avoid large oligomannosyl clusters that may create neo-epitopes, as posited previously with oligomannosides conjugated to virus-like particles[22]. Low-density conjugation also allows better batch-to-batch reproducibility[23]. MALDI-TOF analyses revealed that, on average, 2–5 glycans were conjugated per BSA molecule (Supplementary Figs 27–32).

We first screened all neoglycoconjugates by ELISA with a selection of bnAbs from the PGT128 family (PGT125, 126, and 128) and PGT130 family (PGT130), as these antibodies exhibit varying levels of dependence on the oligomannose at Asn332 and proximal oligomannose arrangements for binding[24]. All antibodies were tested as IgGs. We observed that the three PGT128 family antibodies, particularly PGT128, bound avidly to the NIT82B conjugate (Fig. 3a), which carries an Rv3 derivative with a 1-6-extended D3-like arm. PGT130 bound somewhat less well, which perhaps reflects the alternate manner that this antibody engages oligomannose compared to antibodies of the PGT128 family[24]. The remaining conjugates were bound relatively poorly or not at all by the antibodies.

To investigate these interactions further and determine the extent to which valency of the mimetics on BSA might be important for antibody binding, we performed ELISA inhibition assays with soluble glycosides containing the D3-arm-like extensions and NIT82B as the solid-phase antigen. We found that none of the soluble glycosides inhibited antibody binding to NIT82B as efficiently as the NIT82B conjugate itself (Fig. 3b), suggesting that the antibodies, as IgG, are interacting bivalently with the glycosides on NIT82B. We observed also that binding of all four antibodies was inhibited by both anomeric glycosides with the 1-6-linked extension (NIT68A and NIT68B), although no antibody binding to the α-glycoside was observed when conjugated to BSA (as NIT82A). We reasoned that lack of antibody binding to NIT82A was due to lower average density of the α-glycoside on this conjugate (average 2.3 ligands per mol BSA) compared to the NIT82B conjugate with the β-glycoside, to which the antibodies did bind (average 4.4 ligands per mol BSA) (Fig. 3a). In accordance with this notion, we observed a substantial increase in binding of all four PGT antibodies to a second NIT82A conjugate with a higher ligand density (average 5.4 ligands per mol BSA) and notable reductions in antibody binding to NIT82B conjugates with average ligand densities of 1.5 and 2.6 per BSA (Supplementary Fig. 33). Similarly, we observed that the glycoside NIT70A, with the 1-4-linked D3-like extension, blocked PGT125 and PGT126 binding (Fig. 3b), even though no

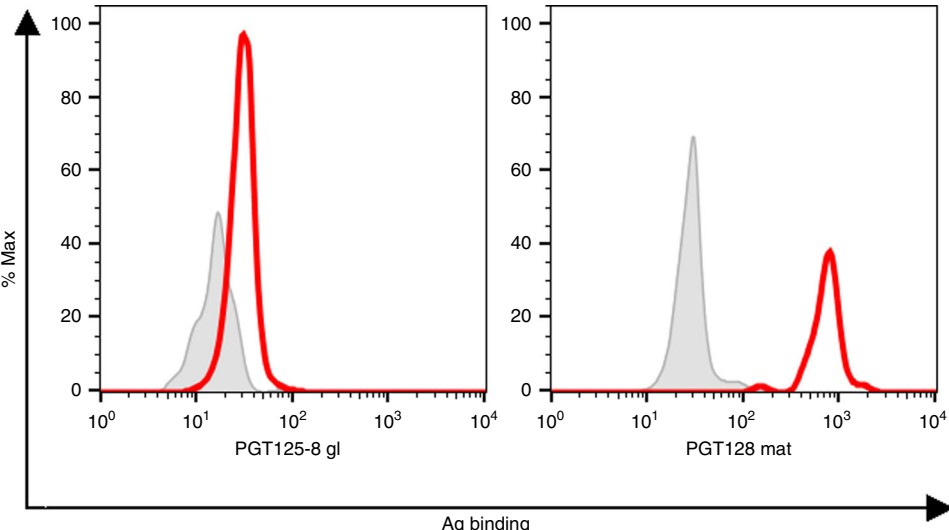

**Fig. 4** The gl precursor of the PGT128/130 families binds the lead NIT82B glycoconjugate. Binding of the predicted gl antibody was determined by flow cytometry using a biotinylated version of the lead conjugate NIT82B (designated NIT150b). Shown is the binding of the gl precursor (left) and of the affinity-matured PGT128 antibody (right), as a control, to NIT150b at 1 μM. Bound antigen was detected with PE-conjugated streptavidin and antigen binding gated on the live, antibody-expressing cell populations, which were detected with an APC-conjugated anti-human F(ab')₂-specific antibody. The greyed histograms represent glycoconjugate binding to cells transfected with the HIV bnAb VRC01, which is specific for the CD4-binding site of HIV Env. The results are from a single experiment

binding to the corresponding glycoconjugate (NIT83A), with on average 3.4 ligands/mol BSA, was observed. Contrary to PGT125 and PGT126, PGT128, and PGT130 binding was not inhibited by the NIT70A glycoside, supporting the notion that they interact with oligomannose in a manner that is different from PGT125 and PGT126[24, 25].

Taken together, the results from our binding and inhibition analyses suggest that an average loading density of 4–5 glycosides per BSA molecule may be needed to obtain ligand spacing on BSA that affords adequate antibody binding avidity. However, it is important to note that we did observe a reproducible difference in the NMR spectra of the α- and β-anomers (Supplementary Figs 5–26), with the $CH_2N_3$ group of the β-anomers showing a triplet signal (denoting that it is freely rotating), but the corresponding $CH_2$ group in the α-compounds showing a complex multiplet (indicating limited flexibility). These observations suggest that the relatively greater flexibility of the β-anomers, in addition to ligand density, may influence antibody binding. Indeed, we consistently observed equal or greater PGT antibody binding to NIT82B conjugates than to NIT82A conjugates when the latter was loaded at comparable or even higher average glycoside density (Fig. 3a and Supplementary Fig. 33). However, whether there is corresponding flexibility when the glycosides are conjugated to BSA remains to be determined.

Finally, we also assayed binding of the lead glycoconjugate to the putative gl precursor antibody of the PGT128 and PGT130 families[20], which we expressed as membrane-bound IgM on the surface of cells to mimic Ig receptors on naive B cells and serve as a proxy for the ability of the conjugate to prime prospective naive B cells that might produce glycan-specific nAbs. Encouragingly, we observed that the gl precursor bound the conjugate with appreciable strength at the assayed concentration (1 μM) (Fig. 4), which has been determined as adequate for engaging naive Ig precursors on B cells in vivo[26]. Thus, our in vitro analyses revealed not only that several oligomannose-specific nAbs were able to bind the lead glycoconjugate avidly, but that the gl precursor also showed binding that should be sufficient for effective engagement of naive B cell receptors upon immunization.

**PGT128 recognition of the lead oligomannose mimetic.** From the results above it was clear that neoglycosides with the 1-6-linked D3-arm extension were reasonable mimics of oligomannose. We therefore determined the crystal structure of PGT128 in complex with glycoside NIT68A, the α anomer of synthetic Rv3 with the 1-6-linked D3-like arm extension, at 2.27 Å resolution to gain insight into the full extent and nature of this mimicry. NIT68A was chosen because it bound PGT128 somewhat better than NIT68B (Fig. 3b). We found NIT68A to be well ordered in the crystal structure (Fig. 5a), except for the linker, and convincing electron density is present for both branches, including the entire D3-like arm extension (Fig. 5b). The unit cell parameters and crystallographic packing of the structure are similar to those of the previously determined PGT128/Man₉ oligomannose complex[21], which facilitates straightforward comparison of the ligand-binding sites. The structures are notably similar in the equivalent portions of the ligands and in the Fab backbone conformation (Fig. 5c). Superimposition of the putative common ligand moieties gives a root mean squared deviation (RMSD) for all atoms of 0.22 Å (0.20 Å for the D3-like arm alone), emphasizing the strong resemblance between the oligomannose mimetic and Man₉ in the context of PGT128 binding. In addition, remarkable conservation of hydrogen bonds observed in PGT128 binding to NIT68A (Supplementary Table 3) further supports the antigenic mimicry of mammalian oligomannose by NIT68A.

**Priming with an oligomannose mimic evokes HIV nAbs.** Given the favorable antigenic qualities of the lead NIT82B conjugate, we wanted to assess its immunogenicity. To approximate human-antibody responses, we conducted immunizations in rats transgenic for human antibodies (OmniRat™). This animal model expresses a chimeric human/rat IgH locus of 22 human $V_H$s and all human $D_H$ and $J_H$ segments together with a human light-chain locus of 16 Vλs linked to Jλ and rat Cλ (Supplementary Fig. 34) or 12 human Vκs linked to Jκ and rat Cκ[27]. Overall, OmniRats exhibit normal immune development, gene usage, and physiological levels of serum antibody and respond to immunogenic stimuli equally as wild-type rats[27, 28].

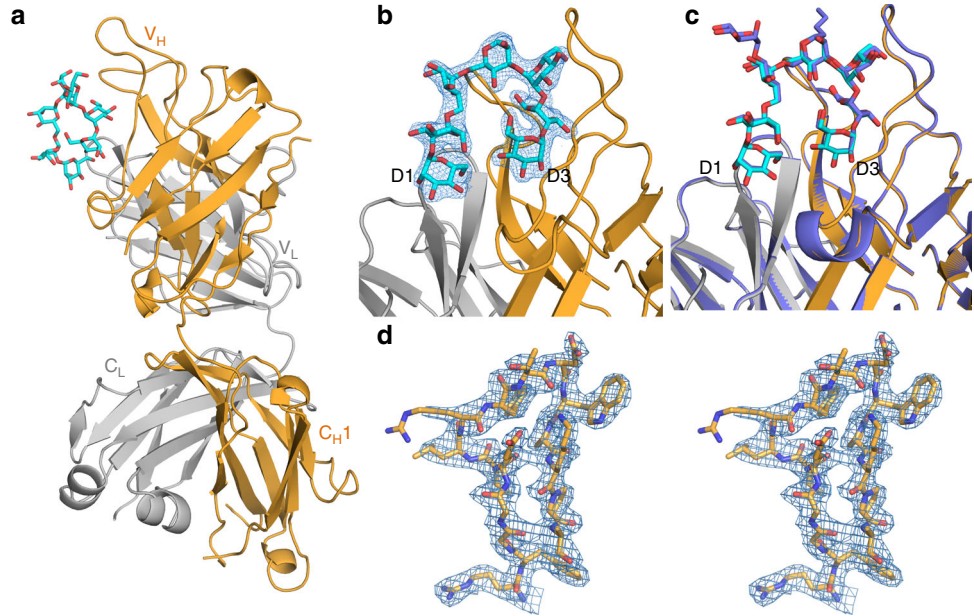

**Fig. 5** Crystal structure of PGT128 with an oligomannose mimetic. **a** Overview of the crystal structure of PGT128 bound to NIT68A. The Fab heavy and light chains are shown in orange and gray, respectively. The NIT68A glycoside is shown in ball-and-stick representation, with carbons in cyan and oxygens in red. **b** View of the NIT68A-binding site of PGT128, with 2Fo-Fc electron density (at $\sigma = 1$) for the mimetic shown as a mesh. **c** View of the NIT68A-binding site of PGT128, colored as above, with the structure of PGT128 in complex with Man$_9$ (PDB 3TV3) superimposed and shown in lavender ribbon and ball-and-stick representation for the respective Fab and ligand. **d** Stereo view of the PGT128 Fab CDRH3, showing the 2Fo-Fc electron density contoured at $\sigma = 1$ for residues 94–101 (shown as sticks)

Three OmniRat animals expressing human λ light chains were primed and then boosted 21 days later with the NIT82B glycoconjugate formulated in the adjuvant Addavax, a commercially available squalene-containing oil-in-water nanoemulsion. Addavax was chosen for these studies because its composition somewhat resembles the proprietary adjuvant MF59, which has been shown to increase immunogenicity of at least some carbohydrate antigens[29]. Pre-immune sera, taken just prior to immunization, and sera collected 6 days after the booster injection at day 21 were assayed.

We first assessed serum binding to a biotinylated derivative of the NIT82B conjugate. We found the immune sera to contain reasonable levels of IgM antibodies; the average $EC_{50}$ titer was 5-fold greater than that of the pre-immune sera (1:40 vs. 1:7; Fig. 6a). IgG binding was unexpectedly meager in the samples from immunized animals. We interpreted these first results as suggestive of T cell independent B cell activation, with predominantly IgM being produced after the booster injection and little IgG. We therefore also assayed the sera for binding to BSA. We found that also the IgM and IgG responses to BSA were not robust (Supplementary Fig. 35). We took from these observations that Addavax had not worked well as an adjuvant for NIT82B; the amount of immunogen seemed a less likely factor, as we had immunized with an amount (30 μg) that normally elicits adequate antibody responses in these animals.

To determine if the elicited IgM antibodies could cross-recognize HIV gp120, we assessed serum binding to virus-dissociated gp120 from three HIV strains captured onto ELISA plate wells. We observed reasonable binding to all three gp120s (Fig. 6b); substantially lower signals were observed with pre-immune sera, indicating that most of the binding of the immune sera was likely due to elicited antibodies and not to naturally occurring ones with specificity for carbohydrate.

We assumed that the gp120 serum binding activity was due to glycan-specific antibodies. To determine if the elicited antibodies were merely a consequence of having immunized with a glycosylated antigen in this animal model, we assayed sera from four OmniRat animals immunized as part of a separate study with recombinant gp120 for binding to the NIT82B glycoconjugate. We observed minimal IgM binding and no IgG binding to NIT82B in comparison to the animals immunized with the glycoconjugate immunogen (Supplementary Fig. 36a). We concluded from these observations that the OmniRat animals immunized with the NIT82B conjugate had not merely generally responded to a glycosylated immunogen, but that the elicited antibodies were in specific response to presented glycosides. An ELISA with recombinant gp120 as solid-phase antigen showed that 3 of 4 gp120-immunized animals contained high levels of anti-gp120 IgG antibodies (Supplementary Fig. 36b), thus demonstrating that the lack of serum binding to the NIT82B conjugate was not due to a general absence of binding antibodies.

Having established that the sera from animals immunized with the glycoconjugate contain binding antibodies with cross-reactivity to HIV gp120, we assessed them for neutralizing activity against a select panel of 7 HIV strains with tier 2-level nAb sensitivity: one subtype A virus (92RW020.2), two subtype B viruses (JRCSF and JRFL), and four subtype C viruses (Du156, Du422, ZM53M, and ZM197M). The viruses were chosen primarily for their variance in sensitivity to bnAbs PGT125-128 and PGT130[24] to allow assessment of the extent to which neutralization patterns of existing nAbs to the oligomannose patch might be mirrored by serum antibodies. Although most of the chosen viruses are highly sensitive to these bnAbs ($IC_{50}$: 0.01–0.2 μg ml$^{-1}$), two are resistant (ZM53M and ZM197M)[24]. We observed that, of the 7 virus strains, 5 were sensitive to serum neutralization ($IC_{50} \geq 1{:}10$), including one of the viruses (ZM197M) that is resistant to PGT125-128 and PGT130 (Fig. 6c). The sera, however, failed to neutralize virus JRFL, which is sensitive to the PGT bnAbs, indicating that the binding modes of the serum antibodies and PGT nAbs differ. No significant neutralizing activity was observed against a negative control virus (vesicular stomatitis virus; VSV) that does not express

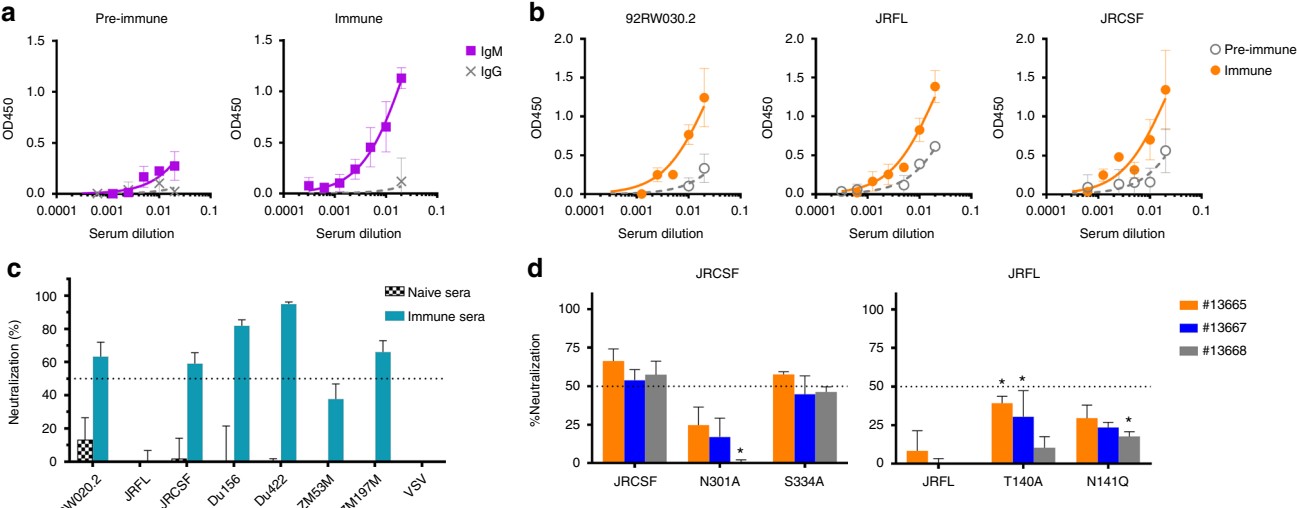

**Fig. 6** Immunization with a bacterial mimic of oligomannose evokes HIV gp120-cross-reactive antibodies with tier 2 neutralizing activity in human-antibody transgenic OmniRats. Animals ($n = 3$) were primed and then boosted (at day 21) with lead conjugate NIT82B formulated in the MF59-like adjuvant Addavax. Sera were collected 6 days thereafter. **a** Binding of IgM and IgG antibody in pre-immune and immune sera to biotinylated NIT82B (5 µg ml$^{-1}$) in ELISA. **b** Comparison of pre-immune and immune serum IgM binding to solubilized HIV gp120 from viruses 92RW020.2, JRFL, and JRCSF. Graphs in **a** and **b** depict mean values for the three serum samples, with error bars denoting the error from the mean. **c** Heat-inactivated sera (1 h, 56 °C) from unimmunized and immunized animals ($n = 3$ each) were assayed (1:10) against a panel of 7 (tier 2) subtype A, B, and C HIV strains and a VSV control virus. Results for the immune sera with the subtype A and B strains are from two independent experiments performed in duplicate or triplicate, while results for the immune sera with the subtype C strains are from a single experiment, performed in duplicate or triplicate. Results for the immune sera with VSV are from seven independent experiments, performed in duplicate or triplicate. Error bars represent the standard error from the mean. **d** Probing of the main serum specificity and factors hindering neutralizing activity. Each bar graph denotes the serum from each of the 3 immunized animals (orange: animal #13665; blue: animal #13667; grey: animal #13668). Left: Comparative assessment of serum neutralizing activity against JRCSF wild-type virus and mutant viruses N301A and S334A. Right: Serum neutralizing activity against JRFL wild-type virus to JRFL virus mutants with glycosylation sites missing in V1 (JRFL-T140A, JRFL-N141Q). The bars denote the mean with SEM from a single experiment performed in triplicate. *$P < 0.05$ (Kruskal–Wallis test with multiple comparisons)

oligomannose on its surface[30] and no neutralizing activity against any of the HIV strains was exhibited by sera from unimmunized animals (Fig. 6c). We inferred from these results that the observed HIV-neutralizing activity was most likely due to the antibodies elicited in response to the glycoconjugate rather than non-specific serum factors.

**The lead mimetic elicits specificities akin to existing nAbs**. We used JRCSF mutant viruses with alanine substitutions at Asn301 (N301A) and Ser334 (S334A), which knocked out the glycans at Asn301 and Asn332 that are usually required for effective binding of oligomannose-specific nAbs[24], to assess whether the serum antibodies were binding to glycans comprising the oligomannose patch on gp120. For the JRCSF-N301A mutant, we observed substantially reduced neutralizing activity relative to wild-type virus with all serum samples, indicating that neutralizing activity was at least somewhat dependent on glycosylation at Asn301 (Fig. 6d, left). In contrast, neutralizing activity against the S334A mutant was unchanged relative to wild-type virus, indicating that the glycan at Asn332 was not essential for serum antibody binding. These specificities mirror those exhibited by some nAbs to the oligomannose patch[24], for example, PGT125 and PGT130, which are highly dependent on the glycan at Asn301 but not on Asn332 for binding. It is important to note that the elicited antibodies may bind to any oligomannose on HIV Env presented in a proper conformation regardless of its position on Env, meaning that the targeted glycans do not all necessarily need be in or around V3. Regrettably, limited serum volumes prevented us from assaying additional virus mutants to further explore serum specificity or possible promiscuity of the serum antibodies in glycan binding.

We were struck by the finding that the immune sera, despite binding to solubilized gp120 from virus JRFL, were unable to neutralize the corresponding virus. JRFL shares > 90% sequence homology to JRCSF, which was somewhat sensitive to serum neutralizing activity. JRCSF and JRFL differ most in the V1 region, with JRFL harboring a third glycosylation site at Asn141 in V1 (Supplementary Fig. 37). Given that glycans in V1 have been shown to modulate access of bnAbs targeting the oligomannose patch around V3[20], we considered that V1 glycosylation in JRFL might be hindering serum nAb activity. We therefore assessed serum neutralizing activity against two JRFL mutant viruses with substitutions that knock out the second or third glycosylation site in V1 (Supplementary Fig. 37), which may be occupied by a mix of oligomannose- and complex-type glycans[31]. We found that both JRFL mutants were notably more sensitive to serum neutralizing activity compared to the wild-type virus (Fig. 6d, right), suggesting that extra glycosylation in the V1 region of JRFL was likely indeed obstructing serum antibody access. However, the resulting neutralizing activity was still less than observed for virus JRCSF, indicating the presence of perhaps additional restrictive factors. Regrettably, these restrictions could not be explored further due to limited serum volumes.

**Elevated antibody levels to oligomannoses in immune sera**. Having observed that the sera exhibit at least some neutralizing activity, we wanted to assess the specificity of the serum antibodies for oligomannose. Sera from each of the three immunized animals plus a pooled pre-immune control sample were therefore submitted to the National Center for Functional Glycomics (NCFG) and assayed on a printed glycan array consisting of 600 glycans in replicates of six (array version 5.3). We focused our

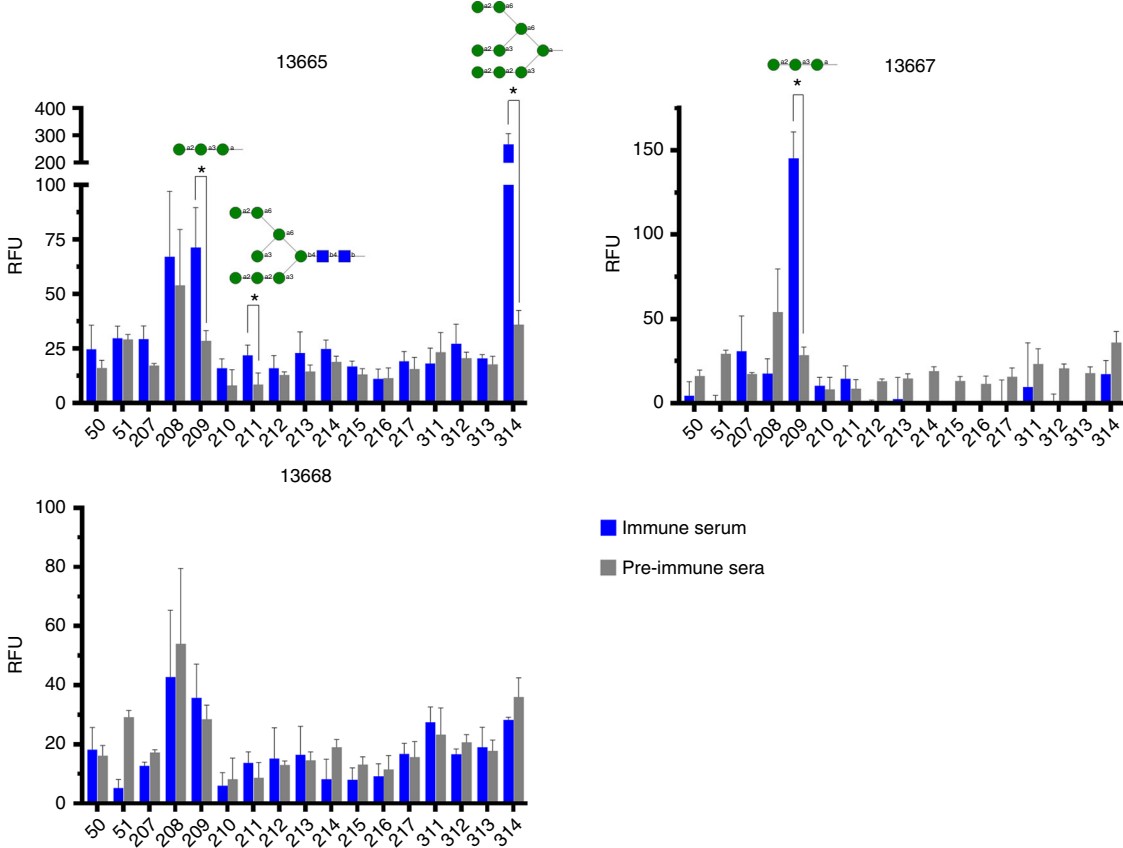

**Fig. 7** Immune sera contain elevated levels of oligomannose-specific antibodies with select specificity as revealed by glycan array analyses. Immune sera ($n = 3$) and a pre-immune control comprised of pooled pre-immune sera from the same three animals were assayed on printed glycan arrays (NCFG; version 5.3) consisting of various synthetic and naturally derived glycosides. Shown is serum IgM antibody detection at a dilution of 1:10 as measured by fluorescence ($y$ axis). The assayed glycans are referenced by their identification numbers on the printed array ($x$ axis). For clarity, only results (from sextuplicate replicates, with outlier signals removed) for high-mannose glycosides and partial structures thereof are shown here. Symbolic representations of the glycosides to which the immune sera bound significantly more strongly than pre-immune control are shown. Statistical significance was determined using the Holm–Sidak method (*$P < 0.05$). Each glycoside was analyzed individually (immune serum vs. pre-immune control), without assuming a consistent standard deviation. The chemical sequences of all arrayed glycosides are listed in Supplementary Table 1

data analyses primarily on oligomannose-type glycans and partial structures thereof. The analyses revealed that two of the sera (from animals 13665 and 13667) contained significantly elevated levels of antibodies with specificity represented by a Manα1-2Manα1-3Man trisaccharide compared to the control sera (Fig. 7). Serum antibodies from animal 13665 with specificity for oligo-Man8 with a chitobiose anchor and synthetic oligo-Man9 were also more abundant than in the pre-immune control. Unexpectedly, no elevated levels of antibodies reactive with oligomannose-type glycans were evident with the serum from animal 13668 (Fig. 7), which we interpreted as reflecting perhaps the presence of antibodies with specificities not represented on the array. Another possibility, which derives from observations with the oligomannose-specific bnAb PGT135[32], is that the serum antibodies from animal 13668 are unable to bind avidly to glycans at the low density that is obtained when printing on the type of glass slide (SCHOTT Nexterion®) used for NCFG arrays.

Serum binding to other glycans on the array was generally not significantly different from the pre-immune control; in instances where binding was significantly different from the control, the difference was often <10-fold (Supplementary Table 2). We did however observe elevated levels of serum antibodies to certain glycosides, albeit with no apparent consistency between the animals. Thus, we found that animal 13665 bound strongly (>30-fold higher than pre-immune control) to a triantennary complex-

type glycan with N-acetyl-lactosamine termini (Supplementary Table 2). Animal 13667 in contrast bound notably stronger to select di- and tetrasaccharides comprised of glucose, N-acetyl-glucosamine or N-acetyl-galactosamine. The third animal (13668) showed no particularly strong binding to any glycan. Because of the inconsistency in binding signals between the animals, it is not clear whether the observed binding is due to varying levels of naturally occurring anti-carbohydrate antibodies in the immunized animals or due to specific antibodies elicited in response to our immunogen.

## Discussion

Glycoconjugate vaccines are a well-established means of eliciting protective antibodies with specificity for glycans on the surface of pathogens. Given that glycans on the surface of HIV Env are known targets for bnAbs during infection, glycoconjugate-based approaches for the elicitation of nAbs to protect against HIV are highly attractive. Here we demonstrate the design and synthesis of a series of related mimetics of oligomannose based on a bacterial oligosaccharide template (Figs. 1, 2 and 5) and show that a single prime plus boost immunization with one of these mimetics, in the form of a glycoconjugate, evokes antibodies that can neutralize a subset of HIV strains (Fig. 6).

Because the predominant response in the sera was IgM (Fig. 6a), we cannot exclude the possibility that at least some of

the observed neutralizing activity was mediated by avidity interactions. Indeed, an engineered IgM form of 2G12 has been shown to neutralize substantially better than monomeric 2G12 IgG[33], which likely results from binding of the multivalent antibody to at least two carbohydrate targets on a single Env spike[34]. We were, however, surprised by the generally low antibody titers following the booster injection and the predominance of IgM antibodies against both the glycans and the protein carrier (Fig. 6a and Supplementary Fig. 35). A few factors may have led to this outcome. First, we know little at this point about which adjuvant might be best suited to promote the development of oligomannose-specific nAbs. The MF59-like adjuvant Addavax was chosen for this study because MF59 has been shown to elicit robust antibody responses with Hib and MenC glycoconjugate antigens[29]. However, results from at least one study show that MF59 can also be a poor adjuvant for some carbohydrate antigens[35]. Addavax may thus not have been the adequate choice for our study, although this will need to be confirmed in future investigations by comparison with other adjuvants. Another factor may be the mimetic itself, which in its current form may not appear sufficiently foreign to the immune system to evoke more robust antibody responses. Indeed, poor antibody titers have been obtained with host-glycan conjugates, for example, with those carrying α2-8-linked poly-sialic acid polymers that are found also on many mammalian tissues[36]. However, chemical engineering to create glycan molecules that are less 'host-like' has been shown to increase immunogenicity[37]. Thus, in addition to assessing other adjuvants, one potential way to boost antibody titers may be to further modify the mimetics so that they appear somewhat more foreign while retaining some similarity to oligomannose. Such investigations are planned.

There is of course some concern that, if elicited, anti-glycan antibodies could cross-react with shared glyco-epitopes on human glycoproteins, leading to autoimmune disorders. At least some of this concern is allayed by the observation that antibodies to shared epitopes have been shown to reside in the body naturally without readily causing autoimmune manifestation[38]. Nevertheless, the development of autoreactive antibodies by a vaccine would likely be undesirable. Future studies will thus need to monitor the development of antibodies with autoreactive signatures or possible adverse events following administration of immunogens designed to elicit antibodies aimed at mammalian glycans on the surface of HIV Env.

Although this report shows that an oligomannose mimetic can prime B cells and evoke antibodies with a modicum of cross-neutralizing activity, we recognize that we are still distant from a vaccine candidate. While further injections with the same glycoconjugate would be expected to increase antibody titers overall, it is possible that this would not be sufficient to yield nAbs of PGT128-like breadth that are most desirable. One possible avenue for increasing neutralization breadth is by using native-like recombinant Env trimers[39] or virus-like particles (VLPs)[40] as boosts to spur further affinity maturation of any elicited nAbs. In such a scenario, neoglycoconjugates would be used for priming (one or more times) followed by boosting with recombinant gp140 trimers (one or more) or VLPs (from single virus or mixture). The HIV antigens would need to be appended with or otherwise incorporate one or more T-helper epitopes derived from the glycoconjugate protein carrier to enable cognate memory T and B cell interactions. The use of HIV-specific antigens would ensure also that antibodies evolve specificity for HIV through interaction with virus-specific protein components, for example, segments of the V3 loop with which PGT128 and related bnAbs also interact[41]. Furthermore, antibodies targeting oligomannose-type glycans on Env will likely need to be somewhat promiscuous in their binding preference and able to

accommodate (or avoid) vicinal glycans within the oligomannose patch to be most effective[20, 24]; the clustering and subtle heterogeneity of glycans on Env may be difficult to achieve by glycoconjugate design but is readily attainable with recombinant trimers and VLPs with glycosylation that mirrors 'typical' glycosylation of circulating HIV strains[31]. We hypothesize that somewhat weakly neutralized viruses would constitute the targets against which the elicited nAbs would need to evolve further. The specific number of recombinant trimers or VLPs to be used for booster injections might then be determined by the extent to which the corresponding weakly neutralized viruses reflect variations in glycosylation that influence serum Ab recognition.

The mimetics approach presented here contrasts with approaches based on the notion that the desired nAbs can be triggered if antigens merely present oligomannose-type glycans in the 'proper' configuration. However, results from one of the few studies to report on the immunogenicity of protein scaffolds engrafted with segments of the Env oligomannose patch showed no significant neutralization of a panel of six tier 2-level HIV-1 strains despite the immunogens appearing to have elicited robust anti-glycan antibodies in at least some animals[42]. Moreover, repetitive immunization of macaques over a period of many years with a recombinant gp140 Env was shown in a recent report to yield serum antibodies that could only neutralize virus when generated in a manner such that all glycans on the Env surface are high-mannose[43]. Somewhat greater support for pursuing antigens presenting oligomannose-type glycans only has come from another study[44] showing the elicitation of PGT121-like nAbs after sequential immunization of PGT121gl precursor-transgenic mice with a series of engineered native-like recombinant HIV trimers. However, because of limited B cell diversity in Ig knockin mice, there remains some uncertainty as to the extent to which the reported strategy would be successful in the context of a more complex antibody repertoire. Given the prominence with which glycans decorate HIV Env and the neutralization breadth exhibited by glycan-targeting nAbs, it is prudent to explore alternative strategies that might elicit equivalent nAbs through immunization.

In conclusion, we show here that a glycan mimetic of mammalian oligomannose represents, to the best of our knowledge, a unique approach for the elicitation of oligomannose-specific nAbs to HIV-1. Optimization of the glycan mimetic and its formulation as a conjugate will be required, including the probing of modifications to potentially instill greater 'foreignness' and the testing of more compatible adjuvants. Because BSA is not a clinically acceptable carrier, another priority will be to substitute it with a more suitable carrier. CRM197 would be particularly attractive in this regard because of its known ability to stimulate robust T-follicular helper cell responses, which may be crucial for bnAb development. Furthermore, systematic characterization of the reactivity of the Lys moieties on CRM197 now allows for the preparation of glycoconjugates with improved batch-to-batch reproducibility[45], particularly at densities that mirror those used here with the NIT82B conjugate. In addition to exploring conjugate optimization, investigations into immune conditions or features of the immune repertoire that could be conducive to the development of oligomannose-specific nAbs during natural infection or certain disease states may also be important, to understand how the development of such antibodies may be stimulated by vaccination.

## Methods
**Reagents and general procedures for glycoside synthesis**. All purchased chemicals were used without further purification unless stated otherwise. The promotor BF₃•Et₂O was used as a solution in diethyl ether (≥ 46% as per the manufacturer). Solvents were dried over activated 4 Å (CH₂Cl₂, N,N-

dimethylformamide, pyridine) molecular sieves. Dry MeOH (Merck) and dry THF (Sigma-Aldrich) were purchased. Cation exchange resin DOWEX 50 H⁺ was regenerated by consecutive washing with HCl (3 M), water and dry MeOH. Aqueous solutions of salts were saturated unless stated otherwise. Concentration of organic solutions was performed under reduced pressure < 40 °C. Optical rotations were measured with a Perkin-Elmer 243 B Polarimeter. $[\alpha]_D$[20] values are given in units of $10^{-1}$ deg cm² g⁻¹. Thin layer chromatography was performed on Merck pre-coated plates: generally, on $5 \times 10$ cm, layer thickness 0.25 mm, Silica Gel 60 $F_{254}$; alternatively, on HPTLC plates with 2.5 cm concentration zone (Merck). Spots were detected by dipping reagent (anisaldehyde-$H_2SO_4$). For column chromatography silica gel (0.040–0.063 mm) was used. HP-column chromatography was performed on pre-packed columns (YMC-Pack SIL-06, 0.005 mm, $250 \times 10$ mm and $250 \times 20$ mm).

NMR spectra were recorded on a Bruker Avance III 600 instrument (600.22 MHz for ¹H, 150.93 MHz for ¹³C) using standard Bruker NMR software. ¹H spectra were referenced to $\delta = 0$ using the TMS signal for solutions in CDCl₃ and DSS for solutions in D₂O (external calibration to 2,2-dimethyl-2-silapentane-5-sulfonic acid). ¹³C spectra were referenced to 77.00 (CDCl₃) and 67.40 (D₂O, external calibration to 1,4-dioxane) p.p.m. Assignments were based on COSY, HSQC, HMBC, and TOCSY spectra. The ESI-MS data were obtained on a Waters Micromass Q-TOF Ultima Global instrument. The MALDI data were obtained on a Bruker Autoflex MALDI-TOF/TOF instrument using 2,5-dihydroxy acetophenone as matrix. For branched oligomers, residue labeling for NMR assignments was done as illustrated in the Supplementary Methods section.

**General procedure A for global deprotection.** A solution of the protected azide derivative in dry MeOH was stirred with 0.1 M sodium methoxide for 48 h at RT under Ar. DOWEX-50WX8 resin (H⁺ -form) was then added to give pH = 7. The resin was filtered off and the filtrate was co-evaporated with toluene to remove methyl benzoate. The crude product was dissolved in MeOH/H₂O/AcOH (1/1/0.05; 2 ml), 10% Pd/C was added and the suspension was stirred under hydrogen atmosphere for 48 h at RT. The catalyst was removed by filtration over Celite® and the filtrate was purified by gel chromatography (Sephadex LH-20; water/MeOH = 2/1). Product containing fractions were lyophilized to give the corresponding amine as colorless solid. The products are partially present in the AcOH salt form.

**General procedure B for neoglycoconjugate preparation.** The amine ligand (2 µmol; 75 eq) was dissolved in 0.1 M aqu. NaHCO₃ (2 ml) and a solution of thiophosgene was added (2 ml of a 6 mM in CHCl₃; 450 eq). The biphasic mixture was stirred vigorously for 2 h. The organic phase was removed with a pipette and the aqueous phase was extracted with CHCl₃ three times. Traces of CHCl₃ were removed by passing a stream of air through the remaining solution. The resulting aqueous phase was added to a solution of BSA (2 mg) in buffer A (2 ml of 0.3 M NaCl and 0.1 M NaHCO₃) and was stirred for 62 h. The mixture was then dialyzed against water (2 L) three times and the residue was lyophilized to give the product as a colorless powder. The amount of conjugated ligand per BSA was determined using MALDI-TOF mass spectroscopy.

**Glycoconjugate biotinylation.** Conjugate NIT82B (0.6 mg) was dissolved in PBS buffer and NHS-biotin was dissolved in water (17 µl; 10 mM (0.88 mg biotin in 200 µl water)). The solution was stirred for 2 h at 0 °C and was then purified on a spin column following the instructions described in the EZ-link Sulfo-NHS-Biotinylation kit (ThermoFisher). The resulting solution was lyophilized. The MALDI data show an average biotin content of ca. 5.6 per BSA.

**In silico modeling of PGT128:glycoside interaction.** To identify locations on the Rv3 oligosaccharide at which to incorporate D3-like branches, the Rv3 crystal structure[19] was modeled into the binding site of a PGT128 crystal structure[21] using the UCSF Chimera software package[46]. To help guide the modeling, the D1 analog segment of Rv3 (residues E-C-B-D) was superimposed onto the D1 arm of Man₉ in the PGT128:Man₉ complex without any further computational manipulation.

**Expression and purification of PGT128 Fab.** The PGT128 Fab was expressed in FreeStyle™ 293 F cells (Invitrogen; cat. no. R790-07) by transfection at a 1:2 ratio of plasmids encoding light and heavy chain (truncated at Asp^H234), respectively[21]. Supernatants harvested 6 days after transfection were passed over an anti-human lambda affinity matrix (CaptureSelect Fab λ; Life Technologies) equilibrated in Dulbecco's PBS. Fab fragments were eluted with 0.1 M glycine, pH 2.7 and neutralized with 1 M Tris-Cl, pH 9.0 at a volume of 2 ml per 15 ml. Eluted fractions were pooled and buffer exchanged into 20 mM sodium acetate, pH 5.6, and loaded onto a Mono S 10/100 GL column (GE Healthcare). A 0–0.5 M KCl linear gradient was used for elution, and the Fab peak eluted at ≈16 mS cm⁻¹. PGT128 peak fractions were pooled, exchanged into 20 mM Tris-HCl, 150 mM NaCl, pH 8.0 buffer and concentrated to 29.5 mg ml⁻¹ with a 10 kDa ultrafiltration concentrator.

**Cell surface IgM expression of PGT128/130 gl precursor.** To enable cell surface expression of the predicted PGT128/130 gl antibody precursor[20] in its IgM form, the 3′ end of the gene encoding soluble IgM in vector pFUSEss-CHIg-hM*03

(Invivogen) was replaced with that of membrane-bound IgM (mI gM)[47] to yield vector pFUSEss-CHIg-mhM*03v2. The $V_H$ segment of the gl precursor was then subcloned into the resulting vector using as PCR template a plasmid encoding the heavy chain of the antibody (kindly provided by Dennis Burton). The resulting heavy-chain-expressing plasmid was then mixed with the transfection reagent Fugene HD (ThermoFisher) at a 1:1 ratio with a light-chain expression plasmid (also kindly provided by Dennis Burton) and transfected into mycoplasma-free 293T cells (ATCC; cat. no. CRL-11268). Cells were trypsinized (TrypLE Express; ThermoFisher) 3 days post-transfection, and re-suspended in FACS staining buffer (Hank's Buffer Staining Solution (HBSS; Lonza) supplemented with 10% (v/v) FBS (Thermo Fisher). The cells were then incubated with NIT150b, a biotinylated version of the NIT82B glycoconjugate (see above), and then with phycoerythrin-conjugated streptavidin and allophycocyanin-conjugated anti-human F(ab')₂ (both from Jackson ImmunoResearch). All incubation steps were performed for 30 min on ice and samples washed and re-suspended in staining buffer between steps. The data were acquired on a BD FACSJazz and analyzed with FlowJo software (v10; Treestar).

**Crystallization.** NIT68A was dissolved in 20 mM Tris-Cl, 150 mM NaCl, pH 7.4 at 24 mg/ml, and was mixed with PGT128 Fab (29 mg ml⁻¹, in the same buffer) at a 1:10 ratio of PGT128: NIT68A, with a final concentration of 19.4 mg ml⁻¹ PGT128 and 4 mM glycoside. Crystal trials were conducted at 4 °C and 20 °C using the JCSG/IAVI/ TSRI CrystalMation robotic system (Rigaku) at The Scripps Research Institute. Crystals were obtained in several conditions, with the best-diffracting crystals being grown at 20 °C in 0.085 M HEPES pH 7.5, 8.5% isopropanol, 10% ethylene glycol, 15% glycerol, and 17% PEG 4000. Crystals were vitrified by plunging into liquid nitrogen.

**Data collection and refinement.** The data were collected on a Pilatus3 6M detector at APS beamline 23-ID-D, a part of the National Institute of General Medical Sciences and National Cancer Institute (GM/CA) Structural Biology Facility at the Advanced Photon Source Advanced Photon Source, Argonne National Laboratory. The data were processed using HKL-2000, and data collection statistics are summarized in Supplementary Table 4.

Molecular replacement was carried out using Phaser[48] with the light and heavy chains of PGT128 (PDB 3TV3) as the starting model. Strong Fo-Fc density for the glycoside was apparent in the resulting map. Phenix[49] was used for refinement and model building was conducted in Coot[50]. Pdb-care[51, 52] was used to validate carbohydrate geometry, and the structure was validated with MolProbity[53]. The final structure had no Ramachandran outliers and 97.5% of residues fell within the favored regions of the Ramachandran plot. Fitting for superimposition calculations was performed using the McLachlan algorithm, as implemented in the program ProFit (Martin, A.C.R.; http://www.bioinf.org.uk/software/profit/).

**Generation of OmniRat animals and immunizations.** Human-antibody transgenic rats were generated as described elsewhere[27, 28, 54]. These animals have been shown to develop normally; the expressed antibody repertoire largely mirrors that in humans. The animals used for this study, designated OmniRat5Lew x Rat4, harbor 22 human $V_H$ genes, including *IGHV4-39* from which the PGT128 and PGT130 nAbs derive, and a complete lambda light-chain locus that includes the *IGLV2-8* gene from which members of the PGT128 and PGT130 nAb families derive.

OmniRat animals (female; 5–7 weeks of age at the start of immunization) were immunized under contract at Antibody Biosolutions (Sunnyvale, CA). For glycoconjugate immunizations, the NIT82B conjugate (1 ml) was mixed with an equal volume of Addavax (Invivogen) as per manufacturer's instructions and an aliquot of this mixture (0.6 ml) then injected subcutaneously into each of three animals, which is the least possible for properly assessing the reproducibility of antibody titers and functional activity. The amount of injected conjugate corresponded to ~ 3 µg of carbohydrate, which is roughly half the amount of conjugated saccharide component used in several human conjugate vaccines currently. No severe adverse reactions were observed. The immunizations were approved by Simon Fraser University's Animal Care Committee (protocol. no. 1089HS-13). A small bleed was collected from all animals prior to immunization and at 6 days after the booster at day 21. The collected blood was left to clot so that serum could be recovered. Samples were stored at −20 °C. Once thawed, they were kept at 4 °C. As controls for virus-neutralization assays, we also exsanguinated 3 unimmunized animals. Serum was recovered from clotted blood and stored in the same manner as described above. No blinding was done.

As part of a separate study, recombinant gp120_JRFL was produced in CHO-K1 cells and purified as previously described[55]. The gp120 was formulated with Quil A adjuvant and used to immunize four OmniRat animals subcutaneously at days 0, 28 and 84 (20 µg per injection) as done previously with non-transgenic mice[55]. Blood was collected at 2 weeks after the last booster injection and serum recovered and stored as described above. These immunizations were approved by Simon Fraser University's Animal Care Committee under a separate protocol (no. 928HS-09).

**ELISA.** For this study, ELISAs were performed as described previously[56], except that PBS supplemented with 3% casein and 0.02% Tween was used as diluent instead of PBS supplemented with BSA (3%) and Tween (0.02%). To assess antibody binding to BSA neoglycoconjugates, microtiter plate wells were coated directly with the conjugates at $5\,\mu g\,ml^{-1}$ (in PBS). Serially diluted mAbs, as IgGs, were assayed for binding starting at $2\,\mu g\,ml^{-1}$. Serum antibodies were assayed starting at a 1:50 dilution.

For ELISA inhibition assays, mAbs (IgG) were incubated at a fixed concentration (corresponding to an $OD_{450}$ of ~ 2 in the absence of inhibitor) for 1 h at RT with serially titrated soluble glycosides (NIT68A, NIT68B, or NIT70A) or the homologous glycoconjugate (NIT82B). This mixture was then applied to separate ELISA plate wells coated with conjugate NIT82B ($5\,\mu g\,ml^{-1}$ in PBS), allowed to incubate for 1 h, and the procedure then completed as described above for the direct binding ELISA.

To assess serum binding to virus-dissociated gp120, pseudovirus preparations were treated with detergent (Empigen; 1% final concentration) and the liberated gp120 captured onto ELISA plate wells with anti-gp120 antibody D7324 (International Enzymes, Inc.) at a concentration of $5\,\mu g\,ml^{-1}$ (in PBS).

For all of the assays above, affinity-matured antibodies were detected with anti-human IgG Fc-specific antibody (Jackson ImmunoResearch) and nitrophenylphosphate substrate (Sigma) as described previously[56]. Bound gl antibody was detected with horseradish peroxidase (HRP)-conjugated anti-human IgG Fc-specific secondary (Jackson ImmunoResearch). Bound serum antibodies were detected with HRP-conjugated anti-rat IgM secondary antibody (Bethyl Laboratories) or an equal mixture of HRP-conjugated anti-rat IgG1 and anti-rat IgG2b secondary antibodies (Bethyl Laboratories). For HRP-conjugated secondary antibodies, tetramethylbenzidine (TMB; ThermoFisher) was used as substrate and the developing reaction stopped with sulfuric acid (2 M) after ~ 10 min and read immediately at 450 nm.

**Pseudoviruses and neutralization assays.** Neutralization assays with pseudo-typed viruses were performed essentially as described previously[18] using mycoplasma-free U87R5 cells as targets (obtained from the NIH AIDS Reagent Program at NIAID). Plasmids encoding for subtype C HIV strains[57, 58] and plasmid pHEF-VSVG encoding for VSV-G[59] were also obtained from the NIH AIDS Reagent Program at NIAID. Plasmids encoding for gp160 of viruses 92RW020.2, JRFL, JRCSF and JRCSF mutants N301A and S334A were from previous studies[60, 61]. The T140A and N141Q substitutions on the JRFL backbone were introduced by QuikChange mutagenesis (Agilent Technologies) and the mutations confirmed by Sanger sequencing. To produce pseudoviruses, mycoplasma-free 293T cells (ATCC) were transiently transfected with Env-expressing plasmids and plasmid pNL4-3.Luc.R-E- as described previously[62].

For serum neutralization assays, serum aliquots were heat-inactivated for 1 h at 56 °C prior to use. Bright-Glo luciferase substrate (Promega) was used to assess luciferase activity. The percentage of neutralizing activity was determined as decribed[62]. Statistical analyses were performed using the Kruskal–Wallis test (GraphPad Prism, version 7.03), with $P < 0.05$ considered significant.

**Glycan array profiling of serum antibodies.** Pooled pre-immune sera and individual immune serum samples were screened at two dilutions (1:10 and 1:100) for glycan binding antibodies as per previously published protocols[63, 64] at the National Center for Functional Glycomics on version 5.3 of a printed array of 600 glycans in replicates of six. Antibody binding was detected using Alexa488-conjugated anti-Rat IgM (Bethyl Laboratories; $5\,\mu g\,ml^{-1}$). GraphPad Prism software (version 7.03) was used to remove likely outliers from replicate relative fluorescence values (RFU) for each glycan and the average RFU and standard deviation then calculated from the remaining replicates. For statistical comparison between the pre-immune and immune samples, the multiple $t$-test feature with Hoski correction was used (Holm–Sidak method; GraphPad Prism, version 7.03). $P < 0.05$ was considered significant.

**Data availability.** Coordinates and structure factors for the PGT128/NIT68A complex reported in this manuscript are available from the Protein Data Bank (www.rcsb.org) under accession code 6B3D. The complete glycan array data set can be found at www.functionalglycomics.org in the CFG data archive under cfg_rRequest_3388. All the other data supporting the findings of this study are available within the article and the Supplementary Information, or available from the authors upon request.

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

## Acknowledgements

This work was supported by funding from the Canadian Institutes of Health Research (CHR-126629 and IBC-150408 to R.P.), the Canadian HIV Vaccine Initiative (THA-118628 to M. Wainberg (Jewish General Hospital); sub-award to R.P.), the Austrian Science Fund (P26919-N28 to P.K.), the NIH (R01 AI084817 and UM1 AI100663 to I.A. W.) and a Collaboration for AIDS Vaccine Discovery grant (I.A.W.). R.P. is supported also by a salary award from the Michael Smith Foundation for Health Research (no. 5268). We thank Alekhya Josyula for assistance with flow cytometry, Ligand Pharmaceuticals and Open Monoclonal Technology for providing OmniRat animals, John Kenney (Antibody Solutions) for coordinating the immunizations and technical advice, John Mascola for providing, through the NIH AIDS Reagent Program, plasmids encoding the VRC01-mature heavy and light chains, and Dennis Burton and Khoa Le (The Scripps Research Institute) for generously providing PGT antibodies and plasmids expressing the PGT128/130 gl precursor. We also acknowledge the participation of the Protein-Glycan Interaction Resource of the Center for Functional Glycomics (supporting NIH grant R24 GM098791) and the National Center for Functional Glycomics at Harvard Medical School (supporting NIH grant P41 GM103694) for glycan array analyses. Molecular modeling was performed with the UCSF Chimera package developed by the Resource for Biocomputing, Visualization, and Informatics at the University of California, San Francisco (supported by NIH grant P41 GM103311). This research also used resources of the Advanced Photon Source, a U.S. Department of Energy (DOE) Office of Science User Facility operated for the DOE Office of Science by Argonne National Laboratory under contract no. DE-AC02-06CH11357. This is manuscript number 29601 from The Scripps Research Institute.

## Author contributions

R.P. and P.K. designed the research, R.P., N.T., S.M., N.L., D.C., and C.R. performed the research, R.P., N.T., S.M., N.L., D.C., I.A.W. and P.K. analyzed the data resulting from the research, R.P., N.T., S.M., I.A.W., and P.K. wrote the manuscript, and all authors commented on drafts of the manuscript and agreed to the final version.

## Additional information

**Competing interests:** The authors declare no competing financial interests.

