## [Peer Review File · Nature Communications]

Reviewers' comments:

Reviewer #1 (Remarks to the Author):

This manuscript describes the synthesis of a five different oligomannosides (three heptamannosides and two pentamannosides), corresponding to high mannose N-glycan structures, as their aminopropyl glycosides and also synthesis of BSA-conjugates of these with a loading between 2.3 and 4.4 oligosaccharide per protein molecule and their subsequent use in investigation into their interaction with HIV nAbs and mice immunization experiments. Well-established synthetic methods (which are properly referenced), both considering the oligosaccharide as well as the conjugate synthesis, have been used to get to the target structures, so no novelty there, but the syntheses have been well-performed and the intermediates and target structures adequately characterized to afford valuable and essential tools for the immunological investigations, but I'm confused about the different numbering of the target oligosaccharides in the Supplementary Information and in the main manuscript. Why are compounds 24, 26, and 28 from the SI called NIT70A, NIT68A, and NIT68B, respectively, in the main manuscript?

Regarding the immunological investigations I'm surprised about the difference in binding of NIT82A and NIT82B to the nAbs (especially since the crystal structure of the complex shows that that part of the structure is not part of the binding) and think that the explanation given, that the higher loading (2.3 vs 4.4) allows the IgGs to "interact bivalently with the glycosides" (line 152, page 7), is not very convincing and surprised that this was not looked into in more details. Why were not NIT82A conjugates with a higher loading made and tested and why were not the NIT82A conjugate also tested in the immunization experiments to show if this different binding to nAbs also would result in a different immune response? Furthermore, if there is a possible difference in freedom of the presented epitopes (lines 165-169, page 8) and the authors think that this might be important to explain the different binding, why were not various linker moieties investigated when making the conjugates? As a conclusion, an interesting chemical biology approach towards an intriguing target, a HIV vaccine, where well-defined synthetic oligosaccharides have been utilized to give valuable immunological data, however, I feel that some of the conclusions drawn are not substantiated by the primary data obtained, and should either be removed or validated by further experiments before the manuscript can be recommended for publishing.

Reviewer #2 (Remarks to the Author):

The work by Pantophlet et al. represents a logical extension of a previous finding by this group, namely that a bacterial oligosaccharide possessing elements of mammalian high mannose oligosaccharides such as Man-8 and Man-9, binds the anti-gp120 antibody 2G12. Here they take that finding further by synthesizing various carbohydrate structures containing different numbers and linkages of termini homologous to Man-9. Neoglycoconjugates are constructed using BSA and the different synthetic sugars, and each is tested for binding or inhibiting binding to known and well-characterized anti-HIV antibodies of the PGT family using ELISA formats. Their results suggest a requirement for at

least 4 mannose mimics per BSA, and the reducing linkage appears to matter as well. Interestingly they identify a conjugate that not only binds to mature PGT antibodies, but also a gI precursor. This is a most interesting finding in the paper because it lends credence to the notion that oligosaccharide antigens can be recognized by immature antibodies, and may therefore be effective immunogens. The authors provide structural basis for Man9 mimicry by crystallizing a NIT-68A-PGT128 complex. Last immunizations with the most effective conjugate NIT-82 induces IgM that binds conjugates and mannose structures on the CFG glycan arrays, whereas controls do not. Overall this is a very well written and thorough paper. It is a bit disappointing that the authors were unable to take the biological characterization a bit further due to lack of sera availability, but it nonetheless provides a quite compelling finding in this important field.

Comments

The NIT-82 series most closely mimics Man9 D1/D3, so the finding that this conjugate is most effective in binding/blocking to PGT antibodies is very nice. It's curious that the alpha linked anomer was ineffective. Reasonable explanations (density, NMR characterization of linker) for this were offered. I wonder if the authors tried to get higher valency of NIT-82a onto BSA to test this further. This could have interesting implications for the effects of remote stereochemistry on terminal conformations.

Page 14 – with regard to glycan array analyses, the authors state that they only focused on oligomannose-type glycans. But did the sera bind to others (hybrid, or even non-N-linked)? This would be an important finding for inclusion in the results, along with some explanation.

Page 16, line 352-355 – not clear to me what the authors are saying. Do they mean to suggest a prime-boost with neoglycoconjugate followed by gp120 trimer, or vice versa? Some explanation would be helpful.

Lines 359-362, with regard to microheterogeneity, I would offer that chemical synthesis offers a means to precisely control the composition of such mixtures, much more than in vivo or recombinant production.

Fig 3, include amounts/concentrations at which sera were tested.

Fig 3, error bars are missing.

Fig 5, similarity/rmsd's for binding of conjugate versus PGT128-Man9 is remarkable.

Fig 5b, structure – why did the authors choose to crystallize the much less effective NIT-68 conjugate to PGT128 for crystallography? Why not NIT-82? And what about the remainder of what would be the D1 arm? Only two mannose units are shown. I may be missing the structure but I can't find NIT-68 in the paper or Supplemental Information. This should be added in the text.

Reviewer #3 (Remarks to the Author):

In this manuscript, the authors designed an array of bacterially-derived glycan mimetics for the V3-glycan broadly neutralizing antibody (bnAb) class by mutuating and adapting the approach used for the glycan-specific 2G12 bnAb. A lead oligomannose mimetic was selected for immunization based on binding to a selection of V3 glycan bnAbs and the PGT128 germline (gl) antibody. The authors demonstrated binding of the neoglycoconjugate to bnAb PGT128 with similar orientation to Man9, and to its germline precursor (as an IgG in ELISA), and demonstrated elicitation of serum IgM cross-reacting with oligomannose expressed on the HIV gp120 surface in Omnicor rats with modicum HIV-1 neutralizing activity after a prime/boost regimen with the lead oligomannose mimetic administered in Adavax adjuvant.

The authors propose that the engagement of naïve B cells targeting oligomannoses through an engineered bacterially-derived oligomannose mimetic may mobilize a pool of cross-reacting bnAb precursors B cells that can be subsequently boosted to include virus-specific components in their cognate epitope. This approach differs from current strategies in that oligomannoses are currently presented in the context of scaffolds including V3 minimal epitopes or full trimeric Envs. This is a fascinating approach and the authors acknowledge that this body of work is "distant from a vaccine candidate", including uncertainty on selection of correct adjuvant, correct mimetic and the concern of eliciting antibodies mediating immune disorders. Hence, it is difficult to gauge the overall relevance of this approach at this stage.

I have only few suggestions for additional experimentation to better corroborate the conclusions, and some more comments that I feel would improve the manuscript if addressed.

The premises of this body of work are that "oligomannose-type glycans on HIV gp120 are now recognized as major targets of bnAbs" and that B cell tolerance mechanisms may limit the frequency of naïve B cells available for engagement by currently available candidate immunogens aimed at inducing V3 glycan bnAbs.

However, the presented work does not address if antibodies elicited through this approach will be subjected to B cell tolerance in humans and, consequentially, if there is any practical advantage of this approach on this regard.

Secondly, the induced responses appear compatible with (or likely) a T-cell independent B cell activation that lead to IgM production. It would have been informative to analyze the serology after priming as it is unclear if the booster had any effect on antibody titers. More relevant, if this is the case, it is unclear how the proposed strategy will effectively boost and affinity mature the pool of B cells originally mobilized, since secondary activation of T cell-independent memory B cells is regulated by antigen-specific IgG (Obukhanych and Nussenzweig, JEM 203(2):305-310, 2006) and their potential to recognize glycoproteins is

unknown. The manuscript would benefit from investigating (B cell phenotype?) and/or commenting this aspect.

A third general concern is about the specificity of the cross-reacting antibodies elicited through this regimen. The authors demonstrate that some of the V3 glycan bnAbs and PGT128 gl bind to the lead mimetic. However, epitope mapping through neutralization (Fig. 6d) shows various degrees of sensitivity to the N301A mutation across sera and, in two of three rats, the mutation reduces but does not abrogate neutralization. This observation is compatible with the hypothesis that a polyclonal response was elicited against multiple epitopes/regions. Given that there is no HIV-1 gp120 Env protein-specific component in the immunogen, won't the elicited antibodies bind to any oligomannose exposed in a proper conformation regardless of its position on Env? If this is correct, the targeted glycans do not necessarily need be in V3 and the observed specificity may simply be a function of where the majority of recognized glycans happened to be at higher frequency across N-linked glycosylation sites in any given batch of pseudoviruses or Env tested (including, of course, V3). This could complicate the refinement of epitope targeting by boosting with complex antigens presenting more than one epitope on the Env glycoprotein. It would be useful for the readership if the authors could comment on this point.

The ability of sera from immunized rats to neutralize multiple viruses is noteworthy, albeit the serum titer of 1:10 is low. Since the response is IgM it cannot be excluded that this level of neutralization was mediated by avidity. The manuscript would benefit from the isolation of the neutralizing antibodies to test their ability to neutralize as IgG, or at least if the authors could comment on this possibility.

On page 8 (lines 177-178) it is stated that "binding [to PGT128 gl] should be sufficient for effective engagement of naïve B cell receptors in vivo". Granted the use of a hypothetical, given the relevance of the observation, it would be beneficial to calculate the K_d , on-rate and off-rate, and to express the PGT128 gl as a BcR (at least in its IgG form) on cell surface to better support this claim. Also, do the author have an explanation for the observed binding to BSA? For ELISA, a negative control antibody should also be reported.

On the same note, PGT128 and PGT130 families are defined as representative of the V3-glycan class of bnAbs (line 92). However, a large body of work has demonstrated ample grade of promiscuity in N-linked glycan recognition (perhaps best exemplified by the PGT121 family), which makes somehow arbitrary the identification of "prototype" antibodies for this class. I suggest rephrasing the sentence to avoid the term "representative".

For the same reason, to support the generalization of the binding to V3 glycan bnAb germ lines and precursors, the manuscript would greatly benefit from testing additional germ lines or minimally mutated V3-glycan bnAb precursors for their ability to bind to the lead mimetic.

Finally, as the authors explain, a loading density of 4-5 neoglycosides per BSA molecule may be required for binding, and densities of 2-3 neoglycosides per molecule appeared to be not sufficient. This was tested on different neoglycosides: to formally prove this point, it

would be useful to compare the binding ability of the same neoglycosides at different loading densities. One practical concern is how a highly homogeneous density can be reproducibly and consistently achieved across different batches: a comment or plan on this regard should be added.

Herewith we provide a point-by-point response to the referees' comments.

Reviewer #1:

(1) I'm confused about the different numbering of the target oligosaccharides in the Supplementary Information and in the main manuscript. Why are compounds 24, 26, and 28 from the SI called NIT70A, NIT68A, and NIT68B, respectively, in the main manuscript?

Response: We regret this confusion, which was due to our oversight at the time of changing our compound numbering to the compound name designation as used in the main text. We have corrected this oversight in the revised manuscript and endeavored to make sure that the names are written consistently throughout the main text and Supplementary.

(2) Regarding the immunological investigations I'm surprised about the difference in binding of NIT82A and NIT82B to the nAbs (especially since the crystal structure of the complex shows that that part of the structure is not part of the binding) and think that the explanation given, that the higher loading (2.3 vs 4.4) allows the IgGs to "interact bivalently with the glycosides" (line 152, page 7), is not very convincing and surprised that this was not looked into in more details.

(a) Why were not NIT82A conjugates with a higher loading made and tested and why were not the NIT82A conjugate also tested in the immunization experiments to show if this different binding to nAbs also would result in a different immune response?

Response: We regret not to have been more thorough in our initial investigation regarding the lack of binding of the PGT antibodies to the main conjugates. To address this point, we generated NIT82A and NIT82B variants loaded with glycosides at higher and lower densities. As shown in the revised manuscript, we observed an increase in binding of all four PGT antibodies to a NIT82A conjugate with an avg. 5.4 ligands/mol BSA and notable reductions in antibody binding to NIT82B conjugates with average ligand densities of 1.5 and 2.6 per BSA (Supplementary Fig. 7). These results thus support the notion stated in the original manuscript that an average glycoside loading density of 4-5 was needed for avid antibody binding, thus resembling the binding interaction of the PGT128 and PGT130 family of antibodies to gp120 (Pejchal et al. Science 334, 1097-1103; Doores et al. J Virol 89, 1105-1118).

(b) Furthermore, if there is a possible difference in freedom of the presented epitopes (lines 165-169, page 8) and the authors think that this ight be important to explain the different binding, why were not various linker moieties investigated when making the conjugates?

Response: We previously wrote that there was a difference in the NMR spectra of the spacers in the α - and β -anomers. However, this was a misstatement. Rather, a higher flexibility was evident for the beta ligand based on the NMR; only the propyl spacer was attached to the ligand in those measurements. We believe therefore that the relatively higher flexibility of beta should be true essentially regardless of the linker we utilize for conjugation. Based on this notion and given the focus of the work presented in this manuscript, we did not pursue an intensive investigation of other linkers. Preliminary data from further work does suggest however that linker length may be a factor influencing antibody binding and we are following up on these investigations separately as part of further work on this project.

(3) As a conclusion, an interesting chemical biology approach towards an intriguing target, a HIV vaccine, where well-defined synthetic oligosaccharides have been utilized to give valuable immunological data, however, I feel that some of the conclusions drawn are not substantiated by the primary data obtained, and should either be removed or validated by further experiments before the manuscript can be recommended for publishing.

Response: We have made further modifications to the text based on suggestions by Reviewers 2 and 3, as per our responses below. We have been careful to not overstate our conclusions and trust that the revised text is now sufficiently balanced.

Reviewer #2:

(1) The NIT-82 series most closely mimics Man9 D1/D3, so the finding that this conjugate is most effective in binding/blocking to PGT antibodies is very nice. It's curious that the alpha linked anomer was ineffective. Reasonable explanations (density, NMR characterization of linker) for this were offered. I wonder if the authors tried to get higher valency of NIT-82a onto BSA to test this further. This could have interesting implications for the effects of remote stereochemistry on terminal conformations.

Response: This comment is similar to comment 2 by Reviewer 1. We concur with both Reviewers that this is an aspect that we should have explored further in our initial investigations. As part of our revisions to the manuscript, we generated additional conjugates with different loading densities. As noted above and in the revised text, we found that increasing ligand density on the NIT82A conjugate indeed resulted in antibody binding (Supplementary Fig. 7). However, we typically observed better PGT antibody binding to the different NIT82B conjugates than to the NIT82A conjugates when loaded at comparable glycoside density (cf. Fig. 3a and Supplementary Fig. 7). As noted in the revised text, we believe that this difference by the relatively greater flexibility of the β -anomers compared to the α -anomers. Preliminary data suggest that this may be a local effect; using more flexible linkers does not appear to automatically yield conjugates that are bound better by the antibodies (data not shown).

(2) Page 14 – with regard to glycan array analyses, the authors state that they only focused on oligomannose-type glycans. But did the sera bind to others (hybrid, or even non-N-linked)? This would be an important finding for inclusion in the results, along with some explanation.

Response: For our glycan array analyses we focused of course primarily on oligomannose-type glycans (and partial structures thereof) because of the objective of our work. Serum binding to other glycans on the array was generally not significantly different from the pre-immune control. We have added an additional supplementary table (new Supplementary Table 2) showing that in those instances in which binding was statistically different from the control, the difference was often less than 10-fold.

(3) Page 16, line 352-355 – not clear to me what the authors are saying. Do they mean to suggest a prime-boost with neoglycoconjugate followed by gp120 trimer, or vice versa? Some explanation would be helpful.

Response: We regret not being clear on this point. We indeed were suggesting a scenario in which neoglycoconjugates would be used for priming (one or more times) followed by boosting with recombinant gp140 trimers (one or more). The trimers (or VLPs) would need to be appended with or otherwise incorporate one or more T-helper epitopes derived from the glycoconjugate protein carrier to enable cognate memory T and B cell interactions. We have added this wording to the Discussion section to help improve clarity.

(4) Lines 359-362, with regard to microheterogeneity, I would offer that chemical synthesis offers a means to precisely control the composition of such mixtures, much more than in vivo or recombinant production.

Response: We agree that chemical synthesis offers many advantages. The point we were trying to make is that the clustering of glycans and glycan heterogeneity of select sites would be easier to mimic using recombinant gp140 trimers (or VLPs) rather than trying to address by chemical design on a glycoconjugate. We have revised the text to make our point clearer.

(5) Fig 3, include amounts/concentrations at which sera were tested.

Response: Fig. 3 shows the binding of purified mAbs to our initial panel of BSA glycoconjugates. The concentration at which they were assayed is evident from the x axis annotation. In the revised manuscript, we also mention in the Figure legend the (starting) concentration of antibody. (In Fig. 6, where we do show serum binding to antibody, the x axis also denotes the serum dilutions used in the assays).

(6) Fig 3, error bars are missing.

Response: We omitted the error bars in the graphs for Fig. 3 because the level of binding (Fig. 3a) and inhibition (Fig. 3b) is unambiguous. In Fig. 3a, for example, the antibodies mostly bind just the single antigen (NIT82B) and we did not feel that error bars would add any further important information. In contrast, we do show error bars in Fig. 6 in which serum binding is depicted.

(7) Fig 5, similarity/rmsd's for binding of conjugate versus PGT128-Man9 is remarkable.

Response: We too were struck by the remarkably high level of similarity given the missing D2 arm. We believe this provides a strong starting point for our envisioned investigations.

(8) Fig 5b, structure – why did the authors choose to crystallize the much less effective NIT-68 conjugate to PGT128 for crystallography? Why not NIT-82? And what about the remainder of what would be the D1 arm? Only two mannose units are shown. I may be missing the structure but I can't find NIT-68 in the paper or Supplemental Information. This should be added in the text.

Response: As noted also to Reviewer #1 above, we regret not being sufficiently careful to explain our compound name designation. In the revised text, we have endeavored to make clearer how the different compounds relate. NIT68 is the name designation of the soluble glycoside that is conjugated to BSA; the name designation of the conjugate is NIT82. We also make clear in the revised text why NIT68A rather than NIT68B was chosen for crystallography. Regarding the structural details, we urge Reviewer #2 to carefully examine the structure as there are three—not two—mannosyl residues visible for the D1 arm. The second mannosyl residue is twisted relative to the mannosyl residue at the non-reducing end of the molecule.

Reviewer #3:

(1) The premises of this body of work are that “oligomannose-type glycans on HIV gp120 are now recognized as major targets of bnAbs” and that B cell tolerance mechanisms may limit the frequency of naïve B cells available for engagement by currently available candidate immunogens aimed at inducing V3 glycan bnAbs. However, the presented work does not address if antibodies elicited through this approach will be subjected to B cell tolerance in humans and, consequentially, if there is any practical advantage of this approach on this regard.

Response: Reviewer #3 is correct that the work presented in the current manuscript does not directly address the tolerance factor. It should be noted however that any B cell tolerance mechanisms that may play a role in humans will not be addressable until we reach clinical trials. To reach that goal, we will first need to probe the strategy in lab animals. We have been steadily building a case for exploring mimicry as a possible avenue for eliciting anti-glycan antibodies to HIV (Clark et al. Chem Biol 19, 254-263 (2012); Doores et al. J Virol 87, 2234-2241 (2013); Stanfield et al. Glycobiology 25, 412-419 (2015)); this manuscript represents a first step to exploring the potential of our approach in a biological system.

(2) Secondly, the induced responses appear compatible with (or likely) a T-cell independent B cell activation that lead to IgM production. It would have been informative to analyze the serology after priming as it is unclear if the booster had any effect on antibody titers. More relevant, if this is the case, it is unclear how the proposed strategy will effectively boost and affinity mature the pool of B cells originally mobilized, since secondary activation of T cell-independent memory B cells is regulated by antigen-specific IgG (Obukhanych and Nussenzweig, JEM 203(2):305-310, 2006) and their potential to recognize glycoproteins is unknown. The manuscript would benefit from investigating (B cell phenotype?) and/or commenting this aspect.

Response: We too considered the possibility that the response may be T-independent. However, we found that also the IgM and IgG responses to BSA were poor (Supplementary Fig. 9 of revised manuscript). We took from these observations that Addavax was not a good choice of adjuvant for NIT82B, despite reports of it working well with other glycoconjugates. Indeed, results from at least one study (Huang et al. Proc Natl Acad Sci U S A 110, 2517-2522 (2013)) show that MF59 can also be a poor adjuvant for some carbohydrate antigens. We have inserted wording to this effect in the revised manuscript. Although we cannot at this time exclude completely the possibility of a T-independent response, we believe that investigations with other adjuvants will show that Addavax was poor for formulation with the BSA conjugate in this reported study.

(3) A third general concern is about the specificity of the cross-reacting antibodies elicited through this regimen. The authors demonstrate that some of the V3 glycan bnAbs and PGT128 g1 bind to the lead mimetic. However, epitope mapping through neutralization (Fig. 6d) shows various degrees of sensitivity to the N301A mutation across sera and, in two of three rats, the mutation reduces but does not abrogate neutralization. This observation is compatible with the hypothesis that a polyclonal response was elicited against multiple epitopes/regions. Given that there is no HIV-1 gp120 Env protein-specific component in the immunogen, won't the elicited antibodies bind to any oligomannose exposed in a proper conformation regardless of its position on Env? If this is correct, the targeted glycans do not necessarily need be in V3 and the observed specificity may simply be a function of where the majority of recognized glycans happened to be at higher frequency across N-linked glycosylation sites in any given batch of pseudoviruses or Env tested (including, of course, V3). This could complicate the refinement of epitope targeting by boosting with complex antigens presenting more than one epitope on the Env glycoprotein. It would be useful for the readership if the authors could comment on this point.

Response: Reviewer #3 is correct to note that the elicited serum antibodies may not be binding to just those glycans at or close to V3 and we have added further wording to the revised text to highlight this very important point. Obviously, the choice to assay the N301A mutant was because it is involved in the binding of most/all HIV-targeting glycan antibodies; given the limited volume of serum at our disposal, we felt that

testing the sera against this mutant was important. Given that the oligomannose patch on HIV Env contains, by definition, the largest density of oligomannose glycans, it seemed clear that our investigations should be focused on that area of Env. We have added some more text in the Discussion section to illustrate our vision for how boosting with Env trimers or VLPs might work. We do not consider our approach as a “refinement of epitope targeting “. Rather, as Reviewer #3 points out, the serum antibody response is inherently polyclonal. Thus, our goal would be to heighten responses to as many glycan targets as the serum antibodies ‘see’ on Env. We believe that this would also promote at least some level of binding promiscuity, as observed the PGT antibodies.

(4) The ability of sera from immunized rats to neutralize multiple viruses is noteworthy, albeit the serum titer of 1:10 is low. Since the response is IgM it cannot be excluded that this level of neutralization was mediated by avidity. The manuscript would benefit from the isolation of the neutralizing antibodies to test their ability to neutralize as IgG, or at least if the authors could comment on this possibility.

Response: We agree that it would have been interesting to clone antibodies from the immunized animals and determine their neutralizing capacity as IgGs compared to IgM forms. Regrettably, because this study was envisioned as a proof-of-concept, we did not plan on recovering antibodies because we anticipated that further optimization (e.g., carrier protein, adjuvant, and immunization protocol) would be needed. We have added wording to the Discussion section noting, as commented on above by Reviewer #3, that the observed neutralizing activity may be mediated by the ability of the IgM serum antibodies to bind at least bivalently to Env and cite relevant papers on this topic that have been studied by other HIV investigators.

(5) On page 8 (lines 177-178) it is stated that “binding [to PGT128 gl] should be sufficient for effective engagement of naïve B cell receptors in vivo”. Granted the use of a hypothetical, given the relevance of the observation, it would be beneficial to calculate the Kd, on-rate and off-rate, and to express the PGT128 gl as a BcR (at least in its IgG form) on cell surface to better support this claim. Also, do the author have an explanation for the observed binding to BSA? For ELISA, a negative control antibody should also be reported.

Response: We too were not satisfied with the ELISA, including for the concerns raised by Reviewer #3. In its stead, we performed, as suggested also by Reviewer #3, flow cytometric analyses using the gl precursor expressed in IgM form on the surface of cells and measured binding of the lead glycoconjugate. As shown in Fig. 4 of the revised manuscript, we can detect appreciable binding at 1 μ M concentration of antigen. Although we agree that knowing the Kd can sometimes be useful, we did not pursue that aspect further for two reasons. The first is technical: To properly determine the Kd of cell surface-expressed antibody by flow cytometric means, it is often necessary to assay at serial antigen concentrations spanning 1-2 orders of magnitude both above and below the suspected Kd of the antibody. We did not have sufficient NIT150b to pursue such a comprehensive analysis. The second reason is based on recent observations that lack of binding in vitro does not automatically signify lack of B cell binding in vivo. One striking example comes from work by Schief and colleagues, who reported recently that despite the lack of measurable affinity to the inferred germline precursor of bnAb PGT121, a single immunization with a designer antigen elicited epitope-specific antibody responses in PGT121-germline knockin mice (Escolano et al. Cell 166 (2016)). Thus, we felt that knowing that our lead conjugate can bind the germline antibody at a concentration that is of relevance in vivo (at 1 μ M) engenders confidence for immunogenicity studies and that determining the Kd would be of limited value in this regard.

(6) On the same note, PGT128 and PGT130 families are defined as representative of the V3-glycan class of bnAbs (line 92). However, a large body of work has demonstrated ample grade of promiscuity in N-linked glycan recognition (perhaps best exemplified by the PGT121 family), which makes somehow arbitrary the identification of “prototype” antibodies for this class. I suggest rephrasing the sentence to avoid the term “representative”.

Response: We agree that the term is not appropriate and have revised the sentence to read that members of the PGT128 and PGT130 families “bind conserved glycans within the oligomannose patch on HIV-1 and exhibit broad neutralizing activity”.

(7) For the same reason, to support the generalization of the binding to V3 glycan bnAb germlines and precursors, the manuscript would greatly benefit from testing additional germlines or minimally mutated V3-glycan bnAb precursors for their ability to bind to the lead mimetic.

Response: For a few HIV bnAbs it has been possible to infer the putative human germline precursors using next-gen sequencing. However, for most nAbs, the true precursors are not known. In those cases, the germline precursor is simulated by so-called germline-reverted antibodies, which are produced from the deduced germline V genes but with retention of (most) of the mature CDR3 sequences. Because CDR3 loops tend to play an important role in antibody binding, germline-reverted versions of bnAbs are generally not considered reliable substitutions for true germline precursors. Hence, while we agree with Reviewer #3 that it would be highly interesting to test the lead mimetic for binding to germline precursors of bnAbs, we are not aware of ‘true germline’ antibodies reported for bnAbs to the oligomannose patch but shall continue to monitor the literature for reports of such antibodies. To our knowledge, the germline precursor of the PGT128/130 families is the best approximation so far reported for the germline sequence of antibodies to this site.

(8) Finally, as the authors explain, a loading density of 4-5 neoglycosides per BSA molecule may be required for binding, and densities of 2-3 neoglycosides per molecule appeared to be not sufficient. This was tested on different neoglycosides: to formally prove this point, it would be useful to compare the binding ability of the same neoglycosides at different loading densities. One practical concern is how a highly homogeneous density can be reproducibly and consistently achieved across different batches: a comment or plan on this regard should be added.

Response: This critique is similar to comment #2 by Reviewer #1. As noted above, we generated NIT82A and NIT82B variants loaded with higher and lower glycoside densities and observed an increase in binding of all four PGT antibodies to a NIT82A conjugate with an avg. 5.4 ligands/mol BSA and notable reductions in antibody binding to NIT82B conjugates with average ligand densities of 1.5 and 2.6 per BSA (Supplementary Fig. 7). These results thus support our original statement that an average glycoside loading density of 4-5 is needed for avid antibody binding. However, we also found by NMR that the β -anomers exhibited relatively greater flexibility, suggesting that flexibility and ligand density may influence antibody binding. Supporting this notion is our finding of equal or greater PGT antibody binding to NIT82B conjugates than to NIT82A conjugates even when the latter was loaded at twice the average glycoside density (*cf.* Fig. 3a and Supplementary Fig. 7). With regard to the concern about batch reproducibility and consistency, we note that conjugate technology has improved substantially in the last few years. One particular advance is the systematic characterization of the reactivity of reactive moieties (typically Lys) on carrier proteins, which in turn allows for the preparation of glycoconjugates with improved batch-to-batch reproducibility – especially when using 3-6 glycosides. Based on Reviewer #3’s suggestion, we have added text to this effect at the end of the Discussion section of the revised manuscript.

REVIEWERS' COMMENTS:

Reviewer #1 (Remarks to the Author):

I'm happy with the authors response to my comments in my previous review and also with the amendments made and can therefore now recommend the manuscript for publication.

Reviewer #3 (Remarks to the Author):

The authors have adequately addressed my queries.

Herewith we provide a point-by-point response to the referees' comments on our revised manuscript.

Reviewer #1:

I'm happy with the authors response to my comments in my previous review and also with the amendments made and can therefore now recommend the manuscript for publication.

Response: We thank the Reviewer #1 for deeming the revised manuscript acceptable for publication.

Reviewer #2:

No comments provided.

Reviewer #3:

The authors have adequately addressed my queries.

Response: We are pleased that our responses and revisions to the manuscript were deemed as sufficiently addressing the previous comments and concerns.